# The forkhead transcription factor Foxj1 controls vertebrate olfactory cilia biogenesis and sensory neuron differentiation

Dheeraj Rayamajhi[1⊙], Mert Ege[2⊙], Kirill Ukhanov[3⊙], Christa Ringers[2,4], Yiliu Zhang[1], Inyoung Jung[2,5], Percival P. D'Gama[2], Summer Shijia Li[1], Mehmet Ilyas Cosacak[6], Caghan Kizil[7], Hae-Chul Park[5], Emre Yaksi[4,8], Jeffrey R. Martens[3], Steven L. Brody[9], Nathalie Jurisch-Yaksi[2,4‡]*, Sudipto Roy[1,10,11‡]*

1 Institute of Molecular and Cell Biology, Agency for Science, Technology and Research, Singapore, 2 Department of Clinical and Molecular Medicine, Norwegian University of Science and Technology, Trondheim, Norway, 3 Department of Pharmacology and Therapeutics, University of Florida, Gainesville, Florida, United States of America, 4 Kavli Institute for Systems Neuroscience and Centre for Neural Computation, Norwegian University of Science and Technology, Trondheim, Norway, 5 Department of Biomedical Sciences, Korea University, Ansan, South Korea, 6 German Center for Neurodegenerative Diseases (DZNE) Dresden, Helmholtz Association, Dresden, Germany, 7 Department of Neurology and The Taub Institute for Research on Alzheimer's Disease and the Aging Brain, Columbia University Irving Medical Center, New York, New York, United States of America, 8 Koç University Research Center for Translational Medicine, Koç University School of Medicine, Istanbul, Turkey, 9 Department of Medicine, Washington University School of Medicine, St. Louis, Missouri, United States of America, 10 Department of Biological Sciences, National University of Singapore, Singapore, 11 Department of Paediatrics, National University of Singapore, Singapore

⊙ These authors contributed equally to this work.
‡ These authors jointly supervised the work.
* nathalie.jurisch-yaksi@ntnu.no (NJ-Y); sudipto@imcb.a-star.edu.sg (SR)

**Data Availability Statement:** All relevant data are within the paper and its Supporting Information files. RNA sequencing datasets are available on GEO with the following accession number

## Abstract

In vertebrates, olfactory receptors localize on multiple cilia elaborated on dendritic knobs of olfactory sensory neurons (OSNs). Although olfactory cilia dysfunction can cause anosmia, how their differentiation is programmed at the transcriptional level has remained largely unexplored. We discovered in zebrafish and mice that Foxj1, a forkhead domain-containing transcription factor traditionally linked with motile cilia biogenesis, is expressed in OSNs and required for olfactory epithelium (OE) formation. In keeping with the immotile nature of olfactory cilia, we observed that ciliary motility genes are repressed in zebrafish, mouse, and human OSNs. Strikingly, we also found that besides ciliogenesis, Foxj1 controls the differentiation of the OSNs themselves by regulating their cell type–specific gene expression, such as that of *olfactory marker protein* (*omp*) involved in odor-evoked signal transduction. In line with this, response to bile acids, odors detected by OMP-positive OSNs, was significantly diminished in *foxj1* mutant zebrafish. Taken together, our findings establish how the canonical Foxj1-mediated motile ciliogenic transcriptional program has been repurposed for the biogenesis of immotile olfactory cilia, as well as for the development of the OSNs.

GSE232397. All raw files are available in the Mendeley Data at this link https://data.mendeley.com/datasets/2pn963jn6y.

**Funding:** This work was supported by a FRIPRO research grant from The Research Council of Norway to N.J-Y. (314189) and by the Bio and Medical Technology Development Program of the National Research Foundation (NRF) funded by the Korean government (MSIT) (2021M3H9A1097594) to H-C.P. and the Agency for Science Technology and Research (A*STAR) of Singapore (SC15-R0010) to S.R. The funders played no role in the study design, data collection and analysis, decision to publish or preparation of the manuscript.

**Competing interests:** The authors have declared that no competing interests exist.

**Abbreviations:** AFW, artificial fish water; BBS, Bardet–Biedl syndrome; BSA, bovine serum albumin; dpf, days post fertilization; IFT, intraflagellar transport; LMP, low melting point; MCC, motile multiciliated cell; mdG, mediodorsal glomeruli; OB, olfactory bulb; OE, olfactory epithelium; OMP, olfactory marker protein; OSN, olfactory sensory neuron; PBS, phosphate buffer saline; PBSTx, TritonX-100 in phosphate-buffered saline; PFA, paraformaldehyde; pre-hybe, pre-hybridization buffer; RE, respiratory epithelium; sgRNA, single-guide RNA; SV2, synaptic vesicle 2; TEM, transmission electron microscopy; TH, tyrosine hydroxylase; vmG, ventromedial glomeruli; vpG, ventroposterior glomeruli; WISH, whole-mount in situ hybridization.

## Introduction

Olfaction plays important roles in food and mate choice and also in the avoidance of predators, making it a vital sensory modality for preservation and reproduction. The olfactory system consists of a highly specialized epithelium within the nasal cavity that contains odor-responsive neurons called olfactory sensory neurons (OSNs). The OSNs have apical specializations called dendritic knobs, which sprout multiple long cilia that intertwine to form an expanded receptive surface, embedded in nasal mucus [1,2]. This ciliary meshwork is the site for the localization of olfactory receptors, which, on binding to odors, activate signaling that is then transmitted to the olfactory bulb (OB) in the brain for perception [2,3]. In humans, the loss of smell is a characteristic feature of many ciliary disorders [4–6]. Consistent with this, mutation of several ciliary genes in model organisms, like the zebrafish and mouse, have been shown to impair olfactory cilia differentiation and, consequently, compromise the sense of smell (for example, see [7–9]). Interestingly, therapeutic interventions to ameliorate ciliary defects in ciliary disorders currently holds the most promise for anosmia. Since the olfactory epithelium (OE) is exposed to the environment, the ease of its accessibility has been exploited for adenoviral-based gene delivery in clinically relevant mouse models, such as for intraflagellar transport (IFT) 88 mutation or Bardet–Biedl syndrome (BBS), with encouraging results [10–12]. Thus, these breakthroughs offer a tangible prospect of gene therapy for curing smell deficits in ciliopathy patients.

There are several attributes of the olfactory cilia that make them distinct from sensory or motile cilia in other cell types. The 2 distinctive morphological characteristics are (a) they are considerably long and filamentous, reaching lengths of up to 200 μm in some species [13]; and (b) they are produced in multiple numbers, about 10 to 30/cell in mice, arising from multiple basal bodies in the dendritic knob [4,14]. In the mouse and the zebrafish, at the ultrastructural level, cilia of the OSNs stand somewhere in between the conventional immotile primary cilia and the motile cilia: They have a (9+2) axonemal microtubule arrangement (like motile cilia) but lack the dynein arms that confer motility (like primary cilia) [15–17].

As mentioned above, genes that have a generic function in ciliary differentiation, such as those encoding components of the key ciliary transport processes, IFT and the BBSome, are also required for proper formation of olfactory cilia [12,18]. However, despite the importance and clinical significance of the olfactory cilia, how OSNs are transcriptionally programmed to generate this unique cilia-type has not been established. In the worm *Caenorhabditis elegans*, the RFX domain containing transcription factor, Daf-19, is required for olfactory cilia biogenesis [19]. Vertebrate genomes possess multiple paralogs of the Rfx family, and some of them have been linked to primary as well as motile cilia differentiation in specific organs and tissues [20]. In addition, the vertebrates utilize the forkhead transcription factor, Foxj1, for programming cells to generate motile cilia [20–22]. Whether these regulatory proteins are also involved in the generation of the specialized immotile olfactory cilia and contribute to their distinctive ultrastructure has, thus far, remained unexplored.

In this study, we show that in the zebrafish as well as the mouse, *Foxj1* genes are expressed in differentiating OSNs, and animals with loss-of-function mutations exhibit a profound disruption in olfactory ciliogenesis. Through transcriptomics analysis as well as direct visualization of gene expression in the OSNs, we found that target genes regulated by Foxj1 in motile ciliated cells that encode components of the ciliary motility complex are repressed. This finding provides a molecular logic of how Foxj1 could be involved in differentiation of the immotile olfactory cilia. Rather intriguingly, besides affecting ciliary differentiation, loss of Foxj1 also affected the development of the OSNs themselves. Notably in the zebrafish, we were able to identify that expression of *ompb* [23–26] and *cnga4* [27,28], encoding 2 key OSN-specific

proteins involved in olfactory signal transduction, were strongly diminished in the absence of Foxj1 activity. In line with this, *foxj1* mutant zebrafish larvae showed reduced response to bile acid that is detected by the activity of Omp-positive OSNs [7,25,29,30]. Thus, in addition to the previously established role of Foxj1 in motile ciliogenesis, here we describe novel functions for Foxj1 in regulating the differentiation of the immotile olfactory cilia and in controlling the development of the OSNs themselves and, consequently, the OE.

## Results

### Foxj1 is expressed in ciliated OSNs of zebrafish and mice

In our earlier analysis of the transcriptional pathways regulating the generation of the motile multiciliated cells (MCCs), we noted that one of the zebrafish orthologs of mammalian *Foxj1*, *foxj1b*, is expressed robustly in the developing OSNs [31,32]. To examine this aspect in further detail, we used a transgenic gene trap strain, *Gt(foxj1b:GFP)^{tsu10Gt}*, which contains GFP inserted within the first intron of *foxj1b* and faithfully reports the transcription pattern of the gene [33]. Consistent with our earlier observations using mRNA in situ hybridization, robust GFP expression could be detected in the differentiating OSNs of the larvae, colabeled using an antibody against the neuronal RNA binding protein HuC (Fig 1A and 1B). Expression of *foxj1b* in OSNs is retained into adulthood, as shown by robust GFP expression throughout the lamella of the adult OE that contains the OSNs (S1A and S1B Fig) [25]. By contrast, we identified that the paralogous gene, *foxj1a*, is predominantly expressed in the motile cilia bearing MCCs that line the periphery of the larval olfactory cavity (Fig 1C) or the tips of the adult OE, and much less prominently in the OSNs (S1B Fig). For this analysis of *foxj1a* expression, we used a newly generated gene trap line, *Gt(foxj1a:2A-Tag-RFP)*, where Tag-RFP was inserted into the *foxj1a* locus and faithfully recapitulates the endogenous expression pattern of *foxj1a*. MCCs also expressed *foxj1b*, and their cilia are labeled with an antibody against glutamylated-tubulin, one of the building blocks of motile cilia (Fig 1D).

Zebrafish OSNs are thought to largely comprise 2 major subtypes—the ciliated OSNs and the microvillar cells, which are intermingled in the OE [25,30]. To determine the cell type specificity of *foxj1* expression, we examined *foxj1b-GFP* expression with markers of either ciliated (olfactory marker protein (Omp)) and the G protein Gαolf or microvillar OSNs (the channel protein Trpc2b). Immunostaining with Gαolf revealed that the *foxj1b* expressing OSNs extend Gαolf-positive cilia into the nasal cavity (Fig 1E; note that only the base of the cilia are indicated in the projection to highlight the colocalization with GFP). We further confirmed that the GFP-positive cells in *Gt(foxj1b:GFP)* are ciliated OSNs by colocalizing GFP with mCherry expressed under the activity of *omp* regulatory elements using a combination of the *Tg(omp:gal4)* and *Tg(UAS:NTR-mCherry)* transgenes [34,35] (Fig 1F). In parallel, we investigated if the *foxj1b*-expressing OSNs could be microvillous OSNs using embryos transgenic for *trpc2b-gal4*; *UAS-Ntr-mCherry* [36]. We found a mutually exclusive expression pattern of GFP and mCherry, implying that the *foxj1* genes are not expressed in the *trpc2b*-expressing microvillar OSNs but primarily in *omp*-expressing ciliated OSNs (Fig 1G).

GFP expression also filled the afferent axons of the OSNs and could be traced right up to their synaptic arborizations on the glomeruli of the OBs. Interestingly, we did not observe a complete colocalization of GFP and Gαolf signals in the OB of larval zebrafish, suggesting that *foxj1b* is expressed in another OSN subtype (Fig 1H). To better decipher the identity of these additional *foxj1b* expressing OSNs, we stained the larval and adult OBs with the presynaptic marker, synaptic vesicle 2 (SV2), which specifically labels the glomeruli [37–39]. We observed that *foxj1b*-expressing OSNs project to a broader range of glomeruli than Ompb-expressing OSNs, including the mediodorsal glomeruli (mdG), the ventromedial glomeruli (vmG), and

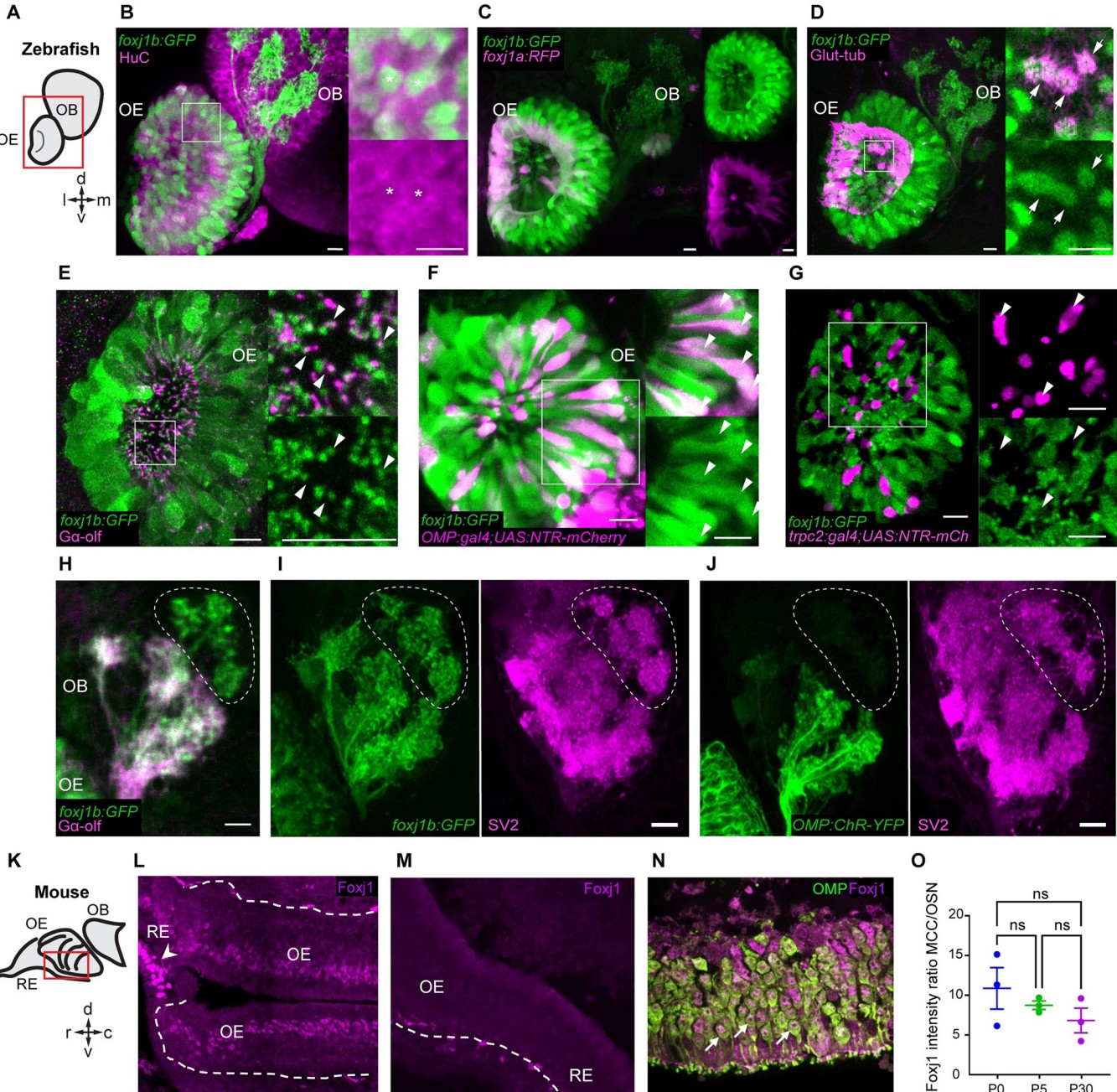

**Fig 1. *foxj1* is expressed in ciliated OSNs of zebrafish and mice.** (**A**) Schematic showing the larval zebrafish OE and OB; l: lateral, m: medial, d: dorsal, v: ventral. (**B**-**J**) Confocal images of the larval (3–5 dpf) zebrafish nose and OB. (**B**) *foxj1b* (*Gt(foxj1b:GFP)*, green) is expressed in neurons labeled by HuC (magenta). Asterisks show *foxj1*-expressing neurons in 5 dpf larva. (**C**) Differential expression of *foxj1a* (*Gt(foxj1a:2A-TagRFP)*, magenta) and *foxj1b* (*Gt (foxj1b:GFP)*, green) paralogs in 5 dpf larvae. Note that *foxj1a* is expressed primarily at the outer rim of the OE. (**D**) *foxj1b*-positive cells (*Gt(foxj1b:GFP)*, green) bear a brush of motile cilia marked by glutamylated-tubulin (magenta) in 5 dpf larvae. (**E**) *foxj1b*-positive OSNs (*Gt(foxj1b:GFP)*, green) in the nasal pit bear cilia, indicated by the marker Gαolf (magenta) in 3 dpf larva. White arrowheads in insets show Gαolf marked cilia arising from *foxj1b*-positive OSNs. (**F**) Overlap of *foxj1b*-expressing cells (*Gt(foxj1b:GFP)*, green) with the ciliated OSN marker *OMP* (*Tg(OMP:gal4); Tg(UAS:NTR-mCherry)*, magenta) in 4 dpf larva. (**G**) Mutually exclusive localization of *foxj1b*-expressing cells (*Gt(foxj1b:GFP)*, green) and microvilli OSNs in 4 dpf larvae, indicated by the marker *trpc2* (*Tg (trpc2:gal4); Tg(UAS:NTR-mCherry)*, magenta). (**H**) *foxj1b*-expressing OSNs (*Gt(foxj1b:GFP)*, green) project mainly to Gαolf-labeled glomeruli (magenta) in 3 dpf larvae. Note that *foxj1b:GFP* and Gαolf do not overlap entirely (dotted region). (**I, J**) *foxj1b*-positive OSNs, in 4 dpf larvae, (**I**) project to more glomeruli than *OMP*-expressing OSNs (**J**) (*Tg(OMP:ChR-YFP)*). Glomeruli are shown by staining with the presynaptic vesicle marker SV2 (magenta). (**K**) Schematic showing mouse RE, OE, and OB; r: rostral, c: caudal, d: dorsal, v: ventral. (**L**) In the nasal epithelium of the adult mouse (P30), Foxj1 localizes to MCCs (arrowhead) within the RE and the layer of OSNs in the OE. Dashed line demarcates the lamina propria separating OE from the underlying tissue. (**M**) Foxj1 is absent from the nasal epithelium of the *Foxj1* knockout mouse (P21). The residual puncta likely represent nonspecific staining of blood vessels or mesenchymal

tissue located in the lamina propria under the basement membrane. (**N**) Foxj1 (magenta) localizes to the nuclei of mature OSNs immunostained for the OMP (P30). (**O**) Expression levels of Foxj1 in OSNs at different ages (newborn P0, day 5 P5, and adult P30) calculated by comparing immunofluorescence in OSNs relative to respiratory MCCs. No significant difference was found (MCC/OSN ratio of 10.87 ± 2.61, P0; 8.74 ± 0.53, P5; 6.83 ± 1.56, P30; fluorescence intensity was measured in 20–30 cells of each type per individual field of view and ratio was calculated per animal, 3 mice in each group) by one-way ANOVA with Kruskal–Wallis test. Scale bars = 10 μm (**B-J**), 50 μm (**L, M**), and 10 μm (**N**). Raw data files are available in Mendeley Data (https://data.mendeley.com/datasets/2pn963jn6y). dpf, days post fertilization; MCC, motile multiciliated cell; OB, olfactory bulb; OE, olfactory epithelium; OMP, olfactory marker protein; OSN, olfactory sensory neuron; RE, respiratory epithelium.

the ventroposterior glomeruli (vpG) (Figs 1I, 1J, S1C and S1D) [40]. We have investigated the identity of this group of cells by screening the existing literature on zebrafish OB glomerular identities. These particular glomeruli have been shown to be negative for the markers for ciliated OSNs (Calretinin and Gαolf) as well as for S100, a marker for crypt cells [38,39]. Crypt, kappe, and pear OSNs are 3 minor kinds of OSNs described in the zebrafish [41–44]. We also found that there was no overlap with the kappe neurons, which have been shown to project to a single target glomerulus among the mdG [42]. Whether the non-Ompb *foxj1b* expressing OSNs are pear cells or a completely different kind of OSN remains to be established.

We also examined Foxj1 expression in the mouse OE. For this, we immunostained the epithelium tissue sections (Fig 1K) with the anti-Foxj1 antibody that specifically recognizes the Foxj1 protein of mammals [45]. We found that MCCs in the respiratory epithelium (RE) were strongly labeled for Foxj1 (Fig 1L), in accordance with previous observations [46]. We also used an antibody against the ciliated OSN marker, OMP, to examine the expression pattern of Foxj1 and OMP in the OE. In the OE of wild-type mice, Foxj1 was detected in mature OSNs as demonstrated by the Foxj1+ signal overlapping with the OMP signal (Fig 1N), albeit at a several fold lower expression level than in the MCCs (Fig 1O). The data aligned well with an earlier report of abundant expression of *Foxj1* mRNA in the adult mouse OE [47]. Importantly, no Foxj1 immunostaining was detected in the OE of the *Foxj1* knockout mice confirming the specificity of the signal produced by the antibody (Fig 1M). Foxj1, being a transcription factor, appeared localized predominantly in the nuclei of the OSNs (Figs 1N (arrows) and S2). We also compared the relative level of Foxj1 expression in the OSNs to that observed in the MCCs and found no significant difference at different postnatal stages of development—P0, P5, and P30 (Figs 1O and S2).

Thus, these findings show specific expression of Foxj1 in ciliated OSNs of both zebrafish and mice.

## Foxj1 regulates olfactory cilia biogenesis in zebrafish and OE establishment in mice

Having established that Foxj1 transcription factors are expressed in vertebrate OSNs, we next examined if they are required for olfactory ciliogenesis. Foxj1 has a central role in motile cilia biogenesis in diverse vertebrates, including the zebrafish, *Xenopus*, mice, and humans [20–22]. Extrapolating from these earlier findings, it seemed reasonable to expect that *foxj1* expression in the OSNs could be linked to the generation of the olfactory cilia. As introduced above, the G-protein Gαolf is enriched on zebrafish and mammalian olfactory cilia and is a useful marker to label these organelles and distinguish them from the motile cilia of the neighboring MCCs. Using Gαolf staining, we detected a reduction in OSN cilia in *foxj1b*, and *foxj1a/b* double mutant, but not *foxj1a* homozygous larvae (Fig 2A–2A'''). Despite the ciliary defects, we did not observe any aberration in the innervation of the Gαolf-positive OSNs to the OB in the double mutant larvae when compared with the control larvae (Fig 2B).

In addition to affecting ciliary differentiation in the OSNs, motile cilia of the MCCs along the lateral edges surrounding the nasal pits [48,49] were also completely absent in *foxj1a/b*

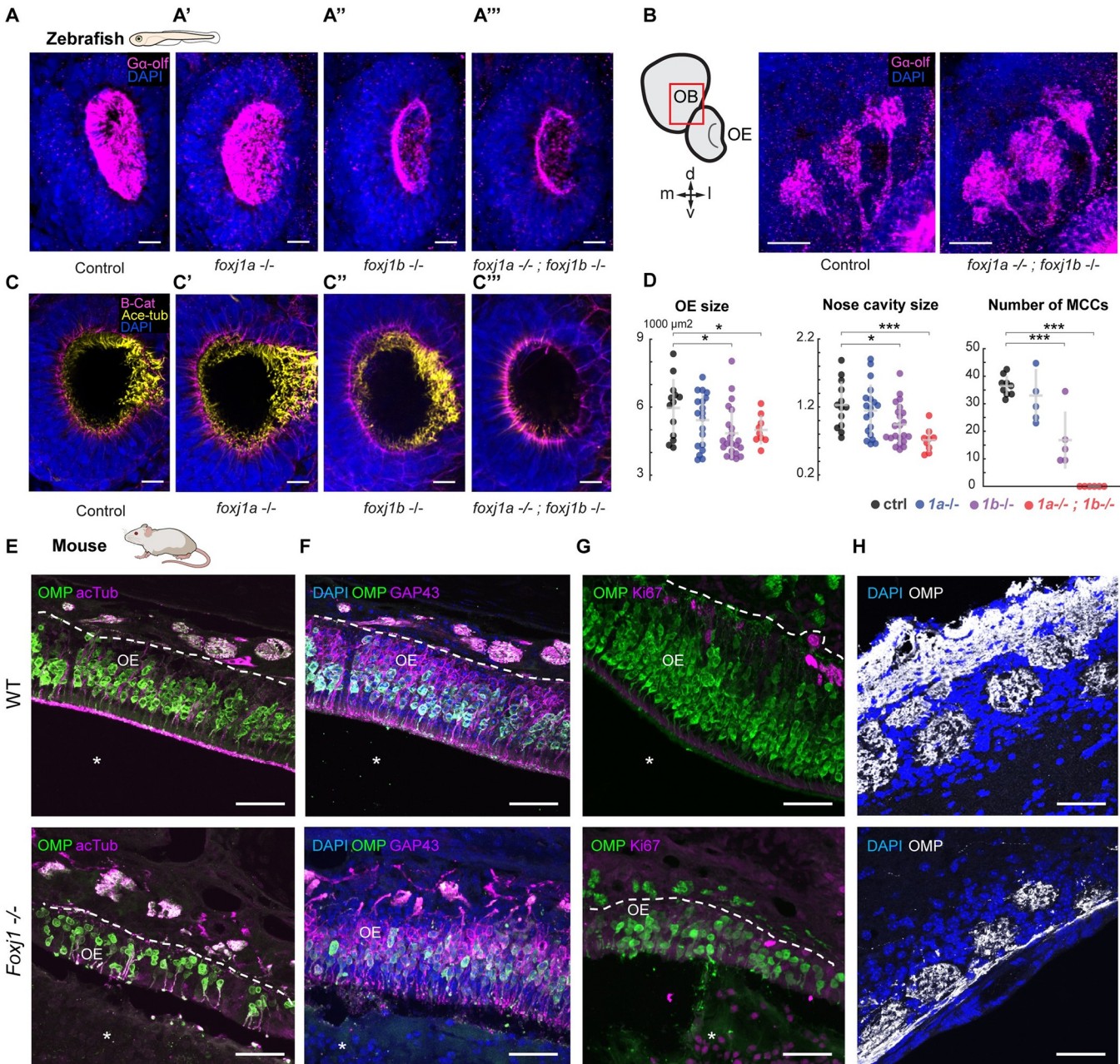

**Fig 2. Foxj1 regulates olfactory cilia biogenesis in zebrafish and OE establishment in mice.** (**A**) Immunostaining with anti-Gαolf antibody marking olfactory cilia (magenta) in *foxj1a,b* mutant zebrafish larvae at 4 dpf. Nuclei marked with DAPI (blue). *foxj1a* mutants showed no observable effect on the formation of olfactory cilia (*n* = 3). *foxj1b* mutants showed reduced olfactory cilia (*n* = 3). *foxj1a/b* double mutants showed severe loss of olfactory cilia (*n* = 3). (**B, B'**) Innervation pattern of ciliated OSNs labeled with Gαolf (magenta) in control (*n* = 3) and *foxj1a/b* double mutant (*n* = 5) showed no significant effect in the formation of olfactory glomeruli at 4 dpf. Nuclei marked with DAPI (blue). (**C**) *foxj1a/b* double mutant showing severe reduction of motile cilia number in comparison to the *foxj1a* mutants, *foxj1b* mutants and control, as shown by immunostaining with anti-acetylated-tubulin for marking cilia (yellow) and beta-catenin for marking cell borders (magenta) at 4 dpf. (nControl = 13, n*foxj1a*−/− = 20, n*foxj1b*−/− = 22, n*foxj1a*−/−; *foxj1b*−/− = 9). Nuclei are marked with DAPI (blue). (**D**) Significant decrease in the size of nose and nasal cavity, and number of MCCs in *foxj1b* and *foxj1a/b* double mutant embryos. (For nose size, nControl = 13, n*foxj1a*−/− = 20, n*foxj1b*−/− = 22, n*foxj1a*−/−; *foxj1b*−/− = 9. For number of MCCs, nControl = 9, n*foxj1a*−/− = 5, n*foxj1b*−/− = 5, n*foxj1a*−/−; *foxj1b*−/− = 6). (**E**) The OE of P21 mouse was stained for OMP (green), a mature OSN marker. Apical layer, composed of mucus and cilia, is strongly labeled for acetylated α-tubulin (magenta) in control (top, arrow). Mature OSNs (OMP, green) at the same animal age of 21 day were fewer in number and disorganized within the OE of the *Foxj1*−/− mouse (bottom), (10,630 ± 923 cells per mm$^2$, *n* = 8, WT; 3,106 ± 714, *n* = 7, KO, 3 mice, *p* = 0.0006). Compared to the WT, thickness of the OE was significantly reduced in the *Foxj1*−/− mutant (79.68 ± 4.31 μm, *n* = 34, 3 mice, WT; 45.48 ± 2.15 μm, *n* = 30, 3 mice, *p* < 0.0001, KO). (**F**) Immature OSNs (GAP43, magenta) were located below the layer of mature OSNs (OMP, green) as shown in a control mouse (top). Immature OSNs (GAP43, magenta) lost their orientation relative to mature OSNs and were fewer in numbers within the OE of the *Foxj1*−/− mutant (bottom)

as compared to control (6,339 ± 1,015 cells per mm$^2$, $n$ = 8, WT; 2,008 ± 481, $n$ = 7, KO, 3 mice, $p$ = 0.0012). The nasal cavity (**E**-**G**, asterisks) was unobstructed in the WT, whereas in the *Foxj1*−/− animals, it was completely filled with DAPI and Ki67-positive cells. (**G**) Proliferating cells expressing Ki67 (magenta) were mostly comprised of basal progenitor cells lining lamina propria. Fewer proliferating cells (Ki67, magenta) were present in the OE of the *Foxj1*−/− mouse (bottom) as compared to the control (top) (2,362 ± 321 cells per mm$^2$, $n$ = 9, WT; 502 ± 72, $n$ = 9, KO, 3 mice, $p$ < 0.0001). (**H**) Images showing the OB having oval-shaped glomeruli (top) filled with the axonal projections of OSNs (OMP, white) in control (top). Reduced intensity of OMP immunostaining in the *Foxj1* −/− mouse (bottom) as compared to the control (top) (352.5 ± 26.7 a.u., $n$ = 21, WT; 266.6 ± 17.5 a.u., $n$ = 44, KO; 3 mice; $p$ = 0.0133). In the *Foxj1*−/− mouse, glomeruli overall were less developed and had smaller perimeter than in the WT (2,558 ± 368 μm, $n$ = 21, 2 mice, WT; 1,331 ± 306 μm, $n$ = 44, 2 mice, KO; 3 mice, $p$ < 0.0001). Scale bars: 10 μm (**A**-**D**), 50 μm (**E**-**H**). Raw data files are available in Mendeley Data (https://data.mendeley.com/datasets/2pn963jn6y). dpf, days post fertilization; KO, knockout; MCC, motile multiciliated cell; OB, olfactory bulb; OE, olfactory epithelium; OMP, olfactory marker protein; OSN, olfactory sensory neuron; WT, wild type.

double mutants (Fig 2C–2C'''), in line with the established function of Foxj1 proteins in motile cilia differentiation. We also identified that the loss of cilia in *foxj1a/b* double mutants coincided with a significant reduction in the size of the nasal cavity (Fig 2D). This suggests a possible impact of Foxj1 mutation on the morphological structuring of the nasal cup or on OSN differentiation (see below and Discussion).

We observed at P21 that loss of *Foxj1* in constitutive knockout mice resulted in a severe disfigurement of the olfactory system at the gross anatomical level; largely undeveloped turbinates were formed, and the entire nasal cavity was filled with DAPI-positive cellular masses composed of neutrophils (S3A, S3A' and S3D Fig), similarly to primary ciliary dyskinesia patients [50]. At the level of the OE, loss of *Foxj1* strongly impacted its cellular composition. In the mutants, the apical ciliary layer of the OE was completely disrupted and largely absent, in part, due to the significant loss of the mature OMP-positive OSNs themselves, while in the OE of wild-type mice, apical cilia appeared as a continuous brush border (Fig 2E). Similar to the mature OSNs, the population of immature GAP43-positive OSNs was also disorganized and greatly depleted in the mutants, while abundant and compactly organized in the wild type (Fig 2F). Loss of both types of OSNs also resulted in a decrease in the thickness of the OE (Fig 2E and 2F). Moreover, loss of Foxj1 resulted in a substantial decrease of active proliferation in the OE as measured by Ki67-positive cells, compared to the wild type (Fig 2G). However, such a profound pathological remodeling of the OE in the Foxj1 knockout mice was not accompanied by cell death as reported by cleaved Caspase (cCas3) immunofluorescence (S3B and S3B' Fig).

Given such a dramatic alteration in the structure of the OE, we asked if the loss of Foxj1 affected the formation of the OB, which strongly depends on the strength of afferent innervation. Overall anatomy of the OB was not significantly affected in the *Foxj1* mutant mice [51]. However, they showed smaller sized glomeruli innervated by OSNs at a significantly lesser density than in the wild type (Figs 2H, S3C and S3C', left panels), as measured by the intensity of OMP immunofluorescence. Smaller size of the glomeruli is most likely linked to a significant reduction in both immature and mature OSNs. Impairment of axonal targeting to the OB was further supported by the existence of wandering axons unable to terminally synapse with glomeruli and stochastically projecting to the inner layers of the OB (S3C' Fig, left-bottom panel (arrows)). This phenomenon is typical of young developing OBs and is not always observed at P21 in the wild-type mouse [52]. Furthermore, tyrosine hydroxylase (TH) immunostaining, a hallmark of normally functioning OB [53], was dramatically reduced in the knockout mice (S3C and S3C', right panels), suggesting a significant decrease of the overall neuronal network activity in the OB.

Taken together, our results show that loss of Foxj1 in zebrafish severely impacts olfactory cilia biogenesis but has lesser impact in OE anatomical organization. On the other hand, Foxj1 loss in mice has severe consequences on the differentiation of the OE and innervation patterns of the OBs.

## Foxj1 controls the expression of OSN-specific genes, but not ciliary motility genes

Foxj1 transcriptional targets include a large set of genes that encode components for motile cilia biogenesis (such as members of the IFT machinery and dynein assembly factors), structural components of the motile ciliary axoneme (such as components of the radial spokes, dynein docking complex, and central pair) and the axonemal dynein proteins themselves, which confer motility [54,55]. Since olfactory cilia are immotile and transmission electron microscopy (TEM) studies have also revealed that their axonemes lack dynein arms [15,17,41,56], we next explored the mechanism by which such a variation to ciliary architecture is brought about, despite being controlled by Foxj1. We analyzed the expression of Foxj1 target genes in single-cell RNA transcriptomes of OSNs obtained from the nasal cavities of wild-type zebrafish, mouse, and human [57–59] and found that there was a notable absence or severe reduction in expression of genes encoding axonemal dyneins and other associated motility components (Figs 3A and S4).

To experimentally validate these observations, we performed whole-mount in situ hybridization in zebrafish embryos for a collection of axonemal dynein genes (*dnah5*, *8*, *9*), as well as *odad1* (aka *ccdc114*, encoding a dynein docking complex component) [60,61] and *ccdc40* (encoding a microtubule binding protein of motile cilia axonemes) [62,63] to document their expression pattern in the MCCs versus the OSNs. In situ hybridization of *dnah5*, *dnah8*, and *dnah9* in 3 dpf *foxj1a/b* embryos revealed all 3 dynein genes to be expressed in the lateral rim of the nasal placodes (the MCC populated region) of control embryos, whereas these genes failed to be expressed in *foxj1a*; *foxj1b* double mutant embryos (Fig 3B), indicating the requirement of Foxj1 for dynein gene expression in the MCCs. Using hybridization chain reaction (HCR) in situ staining, we also colabeled *dnah9* and other 2 genes, *ccdc40* and *odad1*, with *cimap1b* (aka *shippo1*) as well as *ompb*, to check their expression patterns in MCCs and OSNs, respectively. Notably, we found that all 3 motility-related genes expressed in the nasal MCCs (located at the lateral rim of the nasal placode and labeled with the MCC-specific marker *cimap1b*); however, in the OSNs (located within the nasal pit and labeled with the OSN-specific marker *ompb*), we failed to detect their expression (Fig 3C, 3C' and 3C'''). This shows that the OSNs do not express ciliary motility genes.

Next, we examined the impact of loss of Foxj1 on OSN differentiation in zebrafish. We performed RNA sequencing on 4-day-old *foxj1a* and *foxj1b* mutant larvae. We observed that the 2 OSN-specific markers, *ompb* and *cnga4*, were significantly down-regulated in *foxj1b*, but not in *foxj1a* mutants (Fig 3D and 3D' and S1 Table). Omp is a small cytoplasmic protein that is thought to function in modulating odorant response in the OSNs [23–26], while Cnga4, a cyclic nucleotide gated subunit, plays a role in odorant-dependent adaptation [28]. We further examined the status of expression of *ompb* and *cnga4* using whole-mount in situ hybridization in *foxj1a/b* double mutant embryos at 3 dpf. For both genes, we could confirm their dependence on Foxj1 activity: *ompb* and *cnga4* were abundantly expressed in the nose of the control embryos, whereas their expression was almost completely lost in the double mutants (Fig 3E and 3E'). Since *ompb* and *cnga4* encode important components of the olfactory signal transduction machinery, our observations underscore a previously unrecognized function of Foxj1 transcription factors in regulating aspects of the gene expression program specific for OSN maturation.

Based on these observations we conclude that despite the necessity of Foxj1 in the olfactory ciliogenic program, there is a specific repression of its canonical motile ciliary gene targets in the OSNs. Moreover, Foxj1 also controls the differentiation of ciliated OSNs by regulating the OSN-specific gene expression program.

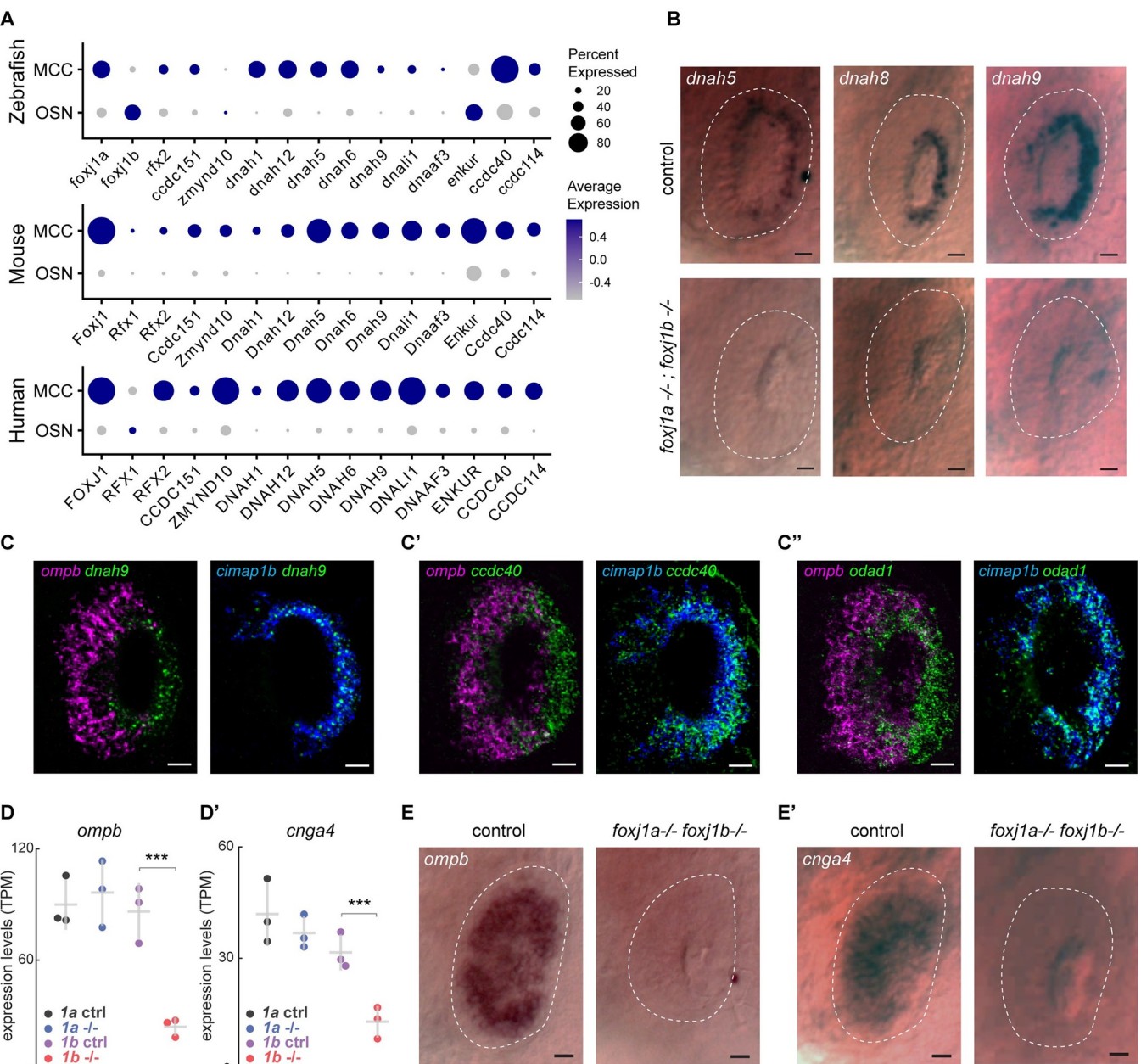

**Fig 3. Foxj1 controls the expression of OSN-specific genes, but not ciliary motility genes.** (**A**) Single-cell transcriptome analysis of ciliated OSNs from wild-type zebrafish, mouse, and human showing low expression of genes encoding axonemal dynein and motility-associated components in OSNs as compared to MCCs. (**B**) Whole-mount in situ hybridization of *dnah5*, *dnah8*, and *dnah9* in *foxj1a,b* double mutant zebrafish embryos at 3 dpf showing absence of gene expression in the periphery of the nasal placodes (highlighted with white dashed line) that primarily consists of OSNs, whereas expression in MCCs at the rim of the cavity (*n* = 10 for each gene). MCCs are depleted in *foxj1a/b* double mutants, and, therefore, expression of these genes were absent in the nasal placodes (bottom panel) (*n* = 5 for *dnah5*, *n* = 6 for *dnah8*, *n* = 4 for *dnah9*). Scale bars = 10 μm. (**C-C"**) Double fluorescent labeling of *dnah9* (**C**), *ccdc40* (**C'**), and *odad1* (**C"**) (green) with ciliated-OSN marker *ompb* (magenta) and MCC marker *cimap1b* (blue) by HCR in situ hybridization in wild-type embryo at 3 dpf showing coexpression of *dnah9*, *ccdc40*, and *odad1* with *cimap1b* (*n* = 3), but not with *ompb* (*n* = 3). (**D, D'**) RNA sequencing of *foxj1a* and *foxj1b* mutant embryos showed a significant decrease in the expression of the ciliated-OSN marker *ompb* (**D**) and *cnga4* (**D'**) in *foxj1b* mutant embryos but not in *foxj1a* mutants. (**E, F**) In situ hybridization in 4 dpf larvae showed reduced expression of *ompb* (**E**) and *cnga4* (**F**) in nasal placodes of *foxj1a/b* double mutant embryos (*n* = 3). Scale Bars = 20 μm (**A-C"**), 10 μm (**E, E'**). Raw data files are available in Mendeley Data (https://data.mendeley.com/datasets/2pn963jn6y). dpf, days post fertilization; HCR, hybridization chain reaction; MCC, motile multiciliated cell; OSN, olfactory sensory neuron.

## Odor responses to bile acids are impaired in *foxj1* mutant zebrafish larvae

As there was a dramatic change in OSN ciliogenesis in the *foxj1* mutants and we could, in addition, identify a role for Foxj1 in OSN differentiation beyond its requirement in ciliogenesis, we finally examined whether olfactory responses are impacted in the absence of its activity. For this, we subjected 4-day-old zebrafish larvae, expressing the calcium indicator GCaMP6 pan-neuronally (using the transgenic line *Tg(elavl3:gcamp6s)*), to various odor types (Fig 4A and 4B). The odor set was composed of mixtures of amino acids, bile acid, nucleic acid, and food odor that activate different subtypes of OSNs and, thus, their corresponding glomeruli

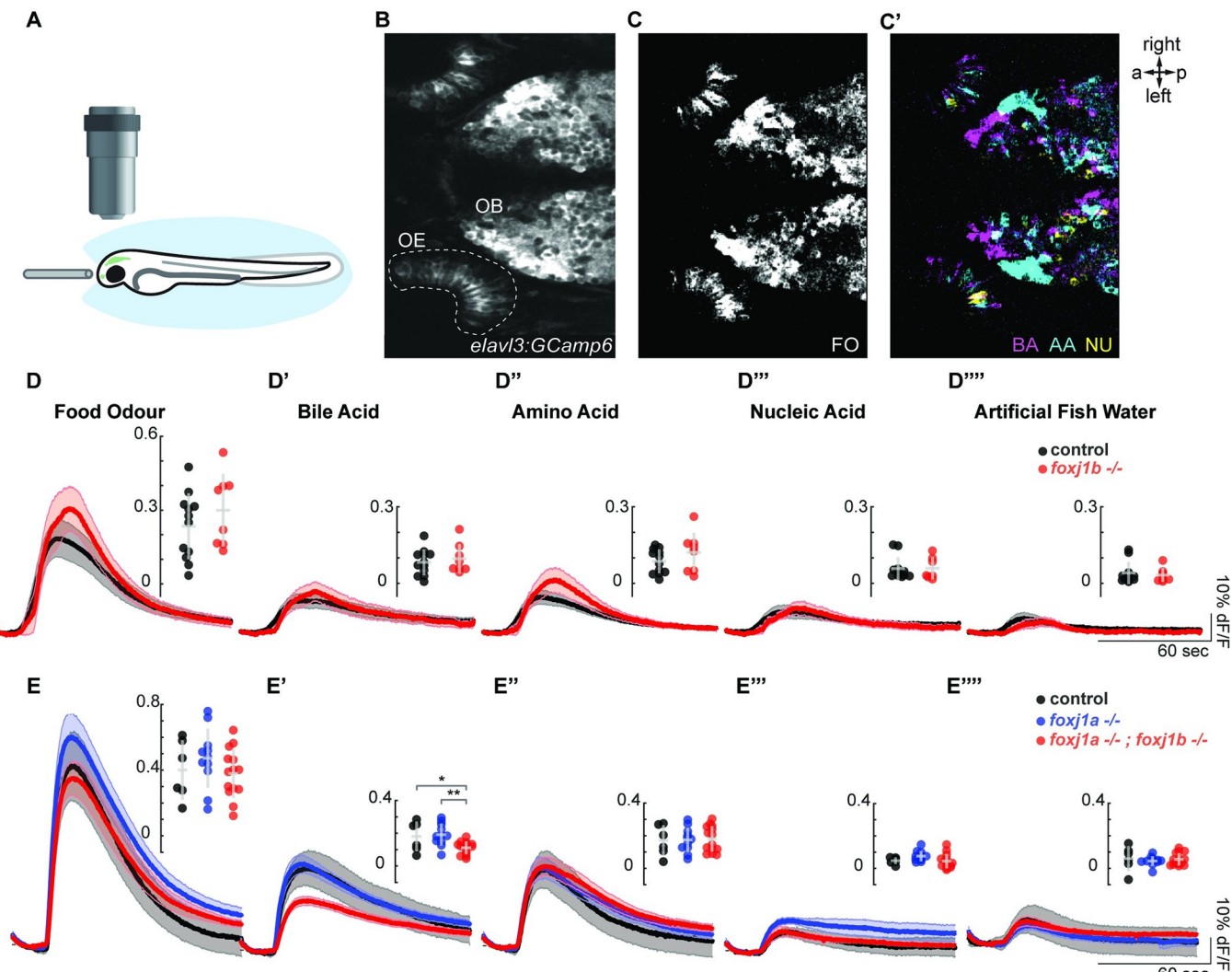

**Fig 4. Olfactory response to bile acid is reduced in *foxj1* mutant zebrafish larvae.** (**A**) Schematic of the olfactory experiment, showing a 4-dpf zebrafish larva embedded in agarose. Its nose was exposed, and odor stimuli were delivered by a fine tube. (**B**) Neural activity was measured using the $Ca^{2+}$-reporter GCaMP6s (*Tg(elavl3:Gcamp6s)*) in a region of interest spanning the entire OE. (**C, C'**) Representative example showing neural activity in the OE and OB to various odor types. Note that responses in the OB were spatially organized, with nonoverlapping domains. (**D-D''''**) Averaged traces of neural activity on the OE for *foxj1b* control ($n = 12$) and mutant ($n = 8$) fish for each odor type showed no difference in odor responses. Maximum amplitude is shown in the insets. (**E-E''''**) Averaged traces of neural activity in the OE for control ($n = 7$), *foxj1a* mutant ($n = 10$), and double *foxj1a/b* mutant ($n = 14$) showed a significant reduction in bile acid response. Shaded error bars are standard error of the mean. Maximum amplitude and standard deviation are shown in the insets. Significance identified by two-sample *t* test, *: $p < 0.05$, **: $p < 0.01$. Raw data files and codes for analysis are available in Mendeley Data (https://data.mendeley.com/datasets/2pn963jn6y). a, anterior; AA, amino acid; BA, bile acid; dpf, days post fertilization; FO, food odor; NU, nucleic acid; OB, olfactory bulb; OE, olfactory epithelium; p, posterior.

[25,29,64,65]. Indeed, by investigating the response patterns in the OB, we were able to show responses to amino acids ventrolaterally, whereas bile acid–responding glomeruli were found medially, as previously described [29,64–66] (Fig 4C and 4C'). Since OSN responses to odor categories are highly selective and the various types of OSNs are intermingled in the epithelium (Figs 1F, 1G and 4C'), we measured the relative change in calcium signal in the entire OE as indicated (Fig 4B) rather than in individual cells. Using these assays, we discovered that loss of function of *foxj1b* alone was not sufficient to impair responses of OSNs to any of the odor categories (Fig 4D–4D''''). However, the combined loss of both *foxj1a* and *foxj1b* led to a significant difference in bile acid response (Fig 4E–E''''). These results suggest that the low expression of *foxj1a* in OSNs (Figs 1C and 3A) and the redundancy between the 2 *foxj1* paralogs prevents the manifestation of a phenotype in the individual *foxj1b* mutant. Since ciliated OSNs are known to be primarily responsible for detecting bile acids, whereas microvilli-containing OSNs detect amino acids and possibly nucleotides [7,25], reduction in bile acid response is in line with our observations of impaired ciliogenesis and differentiation of ciliated OMP-positive OSNs.

## Discussion

We perceive the world through our 5 senses of touch, smell, sight, taste, and hearing. For many of these sensory modalities, cilia play a critical role. For example, photoreceptor neurons elaborate the connecting cilium to transport photopigments, whereas the kinocilium of sensory hair cells of the inner ear is required for proper polarization of the actin-based stereocilia that harbor mechanosensitive channels [67]. Our sense of smell is also dependent on specialized cilia present on the OSNs where odorant receptors are localized [3,67]. Likewise, motility functions of cilia, such as mucus clearance from the airways, circulation of cerebrospinal fluid, and sperm propulsion, also reflect the use of a wide assortment of cilia-types [68]. Understanding how such a great diversity of cilia-types is generated remains a key question in the field.

Ciliation is a rather late event in the development of the OSNs. After their specification, nascent OSNs transit through an immature stage that is characterized by the elaboration of the dendritic knob, generation of multiple basal bodies, migration and docking of these basal bodies with the apical membrane of the dendritic knob, and, finally, extension of the olfactory cilia [3,69,70]. Ciliation happens concurrent with the expression of genes necessary for the OSNs to mature into sensory neurons capable of transducing odor-evoked signals [71–73]. Thus, expression of transduction components like Omp and Cnga4 occur concurrently with the differentiation of the olfactory cilia. Anosmia is a prevalent symptom among many ciliopathy patients, and deficits in the ability to smell can also be inflicted by secondary insults to the olfactory cilia and the OSNs, such as upon viral infections [8,74]. Given these clinically relevant considerations, it is important to understand how olfactory cilia formation is developmentally regulated.

Using genetic analysis, we have now identified that the forkhead domain protein Foxj1, a celebrated "master-regulatory" transcription factor for motile cilia formation, is critical for OSNs to generate their specialized sensory cilia. We also found evidence that the expression of genes for ciliary motility proteins, such as the axonemal dyneins, is suppressed in the OSNs, clarifying why olfactory cilia are immotile and lack dynein arms in electron microscopic studies. In the zebrafish, the levels of *foxj1b* mRNA expression in the OSNs (judged by visual inspection and single-cell RNA seq) is comparable to many cell types that differentiate motile cilia, negating an obvious "levels" issue and leading us to argue that the selective absence of ciliary motility gene expression in the OSNs likely arises from an active transcriptional repression mechanism. However, in the mouse, where we could directly analyze Foxj1 protein expression

in the OE versus the RE, we observed that there is a stark difference, with several fold higher expression in the latter, notably in P0 mice. This observation raises the intriguing possibility that Foxj1 could also be required for earlier stages of development of the OSNs, and not just for their terminal differentiation and ciliation. Clearly, the molecular details underlying the transcriptional activity of Foxj1 in OSNs, particularly the mechanism of repression of cilia motility genes, will need to be elucidated through future investigations. Of note, we have previously shown that *foxj1* is also required in auditory hair cells of the zebrafish ear for their differentiation of the kinocilia [75]. Despite the name (kino = movement in Greek), kinocilia are not actively motile like the prototypical motile cilia. However, hair cells transcribe dynein genes and kinocilia have outer dynein arms but are believed to lack inner dynein arms (and hence the absence of active motility) [76,77]. Taken together, our identification of Foxj1 function in olfactory ciliogenesis further expands the role of this important transcription factor and suggests that its canonical motile cilia-specific transcriptional program is modified in different cells in different ways to generate a diversity of cilia types that are suited for specific purposes. In the context of OSN cilia, we argue that Foxj1 may facilitate their peculiar morphologies such as the 9+2 ultrastructure and extreme length necessary for sensory function. In this connection, it is also important to bear in mind that olfactory cilia in certain amphibian species have been described to be motile [13,17] (see S1 Video for an example of a dissociated OSN from *Rana pipiens*). We speculate that in such instances, Foxj1 likely institutes the characteristic motile ciliogenic transcriptional program in the OSNs to facilitate the formation of motile olfactory cilia. Moreover, while cilia loss has only a mild, yet significant, impact in bile acid responses in the zebrafish as shown in our current study and by Bergboer and colleagues [7], loss of cilia function in mouse or human OSNs is associated with more severe anosmia [4,5,9]. This could arise from the presence of at least 5 different kinds of OSNs in the zebrafish (crypt [41], kappe [42], and pear OSNs [43,44] in addition to the predominant ciliated and microvillar kinds) versus the ciliated OSNs in the OE of mice. Taken together, our findings suggest that even though common principles dictate olfaction across the vertebrates, there are distinct levels of species-specific differences and sensitivity to cilia loss, which may relate to the ecological niches and life-history needs of the individual organisms.

Interestingly, we have also found a more pervasive requirement of Foxj1 in OSN differentiation in the zebrafish as compared to the mouse, which transcends its traditional role in regulating ciliogenesis. In the zebrafish, genes for key olfactory signal transduction proteins, such as *ompb* and *cnga4*, which are expressed during the terminal differentiation process of the OSNs to become functional sensory neurons, are induced by Foxj1. This unexpected finding suggests that Foxj1 has been recruited to the OSN differentiation program independent of its role in regulating ciliary biogenesis. What additional OSN maturation genes are regulated by Foxj1 and whether this function is also relevant in the context of mammals remains to be clarified. In the mouse, in addition to ciliary defects, we observed that loss of Foxj1 had a dramatic consequence on the organization of the OE and innervation patterns of the OBs. In the zebrafish, however, we only observed minor changes in the size of the OE and no major innervation defects. It is important to note that we could only investigate the innervation pattern and size of the OB at larvae stages due to the lethality of the *foxj1a/b* double mutants. Hence, it is still possible that *foxj1* regulates OB development later in development. The difference between zebrafish and mice may also relate to the early expression of Foxj1 during OB neurogenesis in mice [78], but not in the zebrafish. Alternatively, it is possible that motile cilia-mediated flow [48,79], which is also impaired in the *Foxj1* mutant animals, may have a more significant impact on the morphogenesis of the OB in mice than the zebrafish. Further dissection of *Foxj1* function in the OSNs of the mouse will require selective ablation of the gene specifically from these neurons using conditional approaches.

## Materials and methods

### Mouse strains

All procedures performed on mice were approved by the Washington University Institutional Animal Care and Use Committee (IACUC), protocol 22–0117. Knockout *Foxj1* and wild-type littermates were housed in a standard animal facility. Wild-type mice of C57/B6 strain were obtained from the University of Florida mice breeding facility and handled according to the approved IACUC protocol 201908162.

### Zebrafish maintenance and strains

The animal facilities and maintenance of the zebrafish, *Danio rerio*, were approved by the NFSA (Norwegian Food Safety Authority) and the Singapore National Advisory Committee on Laboratory Animal Research. All the procedures were performed on zebrafish larvae of different developmental stages postfertilization in accordance with the European Communities Council Directive, the Norwegian Food Safety Authorities, and the Singapore National Advisory Committee on Laboratory Animal Research. Embryonic, larval, and adult zebrafish were reared according to standard procedures of husbandry at 28.5˚C, unless mentioned otherwise. For our experiments, the following fish lines were used: *T2BGSZ10 Gt(foxj1b:GFP)* [33], *Tg(OMP:Gal4; UAS:NTR-mCherry)* [34,35], *Tg(UAS:GCaMP6s)* [80], *Tg(OMP:ChR2-YFP)* [25], *Tg(trpc2b:gal4)* [36], *Tg(UAS: NTR-mCherry)* [81], *foxj1a* mutants (2 different alleles generated in 2 different laboratories were used [82,83]), *foxj1b* mutant [32], *Gt(foxj1a:2A-TagRFP)^{FRZCC 1100}* (see below), and *Tg(elavl3:GCamp6s)* [84].

Animals were analyzed irrespective of their gender. Note that for zebrafish younger than 2 to 3 months, gender is not apparent. For adult stage imaging experiments, fish were selected according to their body size, which is reported in the manuscript, to ensure reproducibility of the results. Mutants were obtained either from heterozygous incross, heterozygous crossed with homozygous, or homozygous incross. We did not observe an impact of the parents' genotype on the phenotype of the progeny. Controls were siblings with a control genotype (heterozygous or wild type). For homozygous incross, controls were the progeny of a cross of wild-type siblings of the homozygous parents. Animals were genotyped prior to the experiments, and their genotype was reconfirmed following the experiments. Animals were either in the AB or the pigmentless *mitfa*−/− [85] background.

### Generation of *Gt(foxj1a:2A-TagRFP)^{FRZCC 1100}* knock-in zebrafish

To generate the transgenic line to label *foxj1a*-expressing cells specifically, the short homology mediated knock-in strategy was utilized, as previously described [86,87]. First, the Cas9 target site in exon 1 of *foxj1a* gene for the single-guide RNA (sgRNA) was selected using CHOP-CHOP (http://chopchop.cbu.uib.no/). The sgRNA was synthesized by the cloning-free gRNA in vitro synthesis method [86–88] and Cas9 mRNA from the pT3TS-nCas9n vector (Addgene plasmid #46757) with mMESSAGE mMACHINE T3 transcription kit (Thermo Fisher Scientific). To validate the efficiency of the sgRNA, *foxj1a* sgRNA and Cas9 mRNA were injected into one-cell embryos, and genomic DNA was extracted from the injected embryos at 24 hpf. Then, the T7 endonuclease I assay was performed as described in the prior reports [88,89]. Next, the homology arms were designed with the GTagHD (http://www.genesculpt.org/gtaghd/), and the pGTag-TagRFP-SV40 plasmid (Addgene) was used as a donor vector. The *foxj1a* targeting donor vector containing 2A-TagRFP-SV40 [86,87] was constructed, and the plasmid was purified with the QIAGEN Plasmid Midi kit (QIAGEN) and PureYield Plasmid Miniprep System (Promega). Around 2 nl of a mixture of *foxj1a* sgRNA (50 pg), universal

gRNA (25 pg), Cas9 mRNA (200 pg), and the target donor plasmid (10 pg) was microinjected into one-cell stage embryos. Then, the embryos that expressed red fluorescence in the *foxj1a*-expressing cells were screened and raised to adulthood (F0). To identify germline-transmitted lines, the F0 fishes were crossed with wild type and whether the F1 embryos expressed *foxj1a*-specific red fluorescence or not was validated. After the germline-transmitted founder was screened by fluorescence, we confirmed how the DNA is integrated into the genome with the genomic DNAs from the larvae. We verified that the 2A-TagRFP-SV40 DNA is properly integrated into the target site by performing PCR at the 5′ and 3′ junctions of integration sites and sequencing.

## Two-photon calcium imaging

For in vivo imaging, fish were paralyzed through α-bungarotoxin injection [48,90] and were embedded in 0.75% low melting point (LMP) agarose (Thermo Fisher Scientific) in a recording chamber (Fluorodish, World Precision Instruments). To ensure odor delivery to the nostrils, the LMP agarose in front of the nose was carefully removed, after letting solidification for 30 minutes.

A 2-photon microscope were used for calcium imaging: Scientifica, with a Nikon 16× NA 0.8 water immersion objective. For excitation, a mode-locked Ti:Sapphire laser (MaiTai Spectra-Physics) was tuned to 920 nm [90]. Recordings were performed as single plane recordings. The acquisition rate was 31 Hz with an image size of 512 × 510 pixels. Data analysis was done with MATLAB as described in the subsequent section.

## Odor preparation

Our odor selection consisted of food odor, bile acid mixture (taurocholic acid, taurodeoxycholic acid), amino acid mixture (alanine, phenylalanine, aspartic acid, arginine, methionine, asparagine, histidine), nucleic acid mixture (inosine monophosphate, adenosine monophosphate) [7,48,66]. All odorants were purchased from Sigma Aldrich. Food odor was prepared using commercially available fish food; 1 g of food particles was incubated in 50 ml of artificial fish water (AFW) for at least 1 hour, filtered through filter paper, and diluted to 1:50. Amino acid mixture, bile acid mixture, and nucleotide mixture were all prepared from the frozen stocks the day before use at 0.1 mM and were kept at 4°C. All odor mixtures were allowed to equilibrate to room temperature before the experiments.

## Odor delivery

Odors were delivered with a tube positioned right in front of the nose for 20 seconds through a constant flow of AFW (2 ml/min). The stimulation was performed with HPLC injection valve (Valco Instruments) controlled with Arduino Due [7,48,66]. Before each experiment, a trial with fluorescein ($10^{-4}$ M in AFW) was performed to determine precise onset of odor delivery. The 4 odor mixtures, as well as AFW, were presented in a randomized order, and this order of delivery was repeated thrice in total per fish. After the experiment, the larvae were retrieved for health check and genotyping.

## Data analysis

Once the neural activity was recorded, the data were aligned [48]. Distinct ROIs corresponding to the left and right OEs were drawn. The change of fluorescence was estimated as the relative change of fluorescence over time by dF/F, F-$F_{baseline}$/$F_{baseline}$. $F_{baseline}$ was the average value of the frames at the onset of the recording before the stimulus delivery. A baseline of 10 seconds,

approximately 310 frames, was used. The maximum of the response amplitude of the change of fluorescence was identified for each trial using the findpeaks function in the first minute of the recording, and the average maximum was calculated across the left and right for the 3-trial repetitions.

Data analysis and statistics was done using MATLAB. Two-sample *t* test was used for analysis. $P < 0.05$ was considered as statistically significant. All the raw data files and codes for analysis are available in Mendeley Data at this link https://data.mendeley.com/datasets/2pn963jn6y.

## Immunostaining and HE staining of the mouse olfactory system

Adult mice used in this study were anesthetized with ketamine/xylazine and cardiac perfused using ice-cold phosphate buffer saline (PBS) (pH 7.4), immediately followed by ice-cold 4% paraformaldehyde (PFA) in PBS. Air from nasal cavity was removed by vacuum to enhance access of the fixative to the OE. Neonate mice at P0 and P5 were anesthetized on ice, decapitated, and heads drop-fixed in ice-cold fixative. The tissue was postfixed overnight, decalcified in 0.5 M EDTA (pH 8), incubated in 10%, 20%, and 30% sucrose, and frozen in Tissue-Tek OCT embedding medium (Thermo Fisher Scientific, cat# NC1862249). Importantly, immunodetection of Foxj1 required fixation in 1% PFA. Fixed tissue was sectioned at 12 μm thickness, and sections were mounted on a Superfrost glass slides (Thermo Fisher Scientific Cat# 12-550-15). Except for Foxj1, immunostaining of all other marker proteins did not require an antigen retrieval. However, Foxj1 detection in mouse OE required heat-activated retrieval using Tris/EDTA (1 mM EDTA, 0.05% Tween 20 (pH 8.0)) and incubation for 20 minutes at high pressure in a consumer-grade pressure cooker (Instant Pot Duo). For immunostaining, tissue sections were washed $3 \times 5$ minutes in PBS, then permeabilized for 15 minutes in PBS with 0.1% Triton X-100, and, finally, incubated overnight at 4˚C in blocking solution containing 2% normal donkey serum (Jackson Immunoresearch, cat# 017-000-121, RRID: AB_2337258), 0.5% bovine serum albumin (BSA) (Sigma cat# A7030) prepared in PBS and 0.1% Triton X-100. Afterwards, tissue sections were incubated overnight at 4˚C with primary antibodies diluted in the blocking solution. Dilutions of primary antibodies were as follows: anti-Foxj1 (1:100), anti-OMP (1:2,000), anti-acetylated tubulin (1:1,000), anti-Gap43 (1:500), anti-cleaved Caspase3 (1:500), and anti-TH (1:500). Next, tissues were washed $3 \times 5$ minutes with PBS and incubated for 1 hour at room temperature with respective secondary antibodies (1:1,000 dilution). Finally, the tissues were counterstained with DAPI and embedded in Prolonged Gold (Thermo Fisher Scientific cat# P10144). The processed tissues were analyzed with an inverted confocal microscope Nikon TiE-PFS-A1R using preset optical configuration for DAPI, FITC, TRITC, and Cy5. Images were postprocessed using Nikon Elements software (version 5.11) and NIH ImageJ/FIJI.

Chromogenic HE staining of the tissue was done according to the manufacturer's instructions (Vector Laboratories H-3502).

## Analysis of immunofluorescence and morphology of the mouse OE

Number of cells immunostained with a respective marker protein was counted in several representative areas of the OE and then normalized to the area of the OE delineated by the basal lamina and uppermost apical layer. This approach accounted for the difference in the OE thickness in the wild type and *Foxj1*-KO. Intensity of immunostaining for Foxj1 was measured in individual cells identified in the RE and OE, then grouped per type to compute an average and derive the ratio between respiratory MCCs and olfactory OSNs per each representative tissue area. Special care was taken to acquire immunofluorescence images of the wild-type and

*Foxj1*-KO tissue at identical optical settings. Glomeruli in the OB were analyzed by measuring its perimeter whereby immunostaining was measured as an average fluorescence within each glomerulus delineated by DAPI staining. Statistical analysis of the data was done by Prism 9 (GraphPad). Unpaired two-tail Mann–Whitney test was used to compare 2 groups and $p < 0.05$ was considered statistically significant. Three-group comparison was done using one-way ANOVA with Kruskal–Wallis test.

### Immunostaining of larval zebrafish nose

Zebrafish larvae at 4 dpf were fixed with a 4% PFA solution and 0.5% TritonX-100 in phosphate-buffered saline (PBSTx) at 4°C overnight. The samples were washed with 0.5% PBSTx to remove the fixing solution. The larvae were permeabilized in pre-chilled 100% acetone for 20 minutes at −20°C, after which the samples were washed with 0.5% PBSTx (3 × 10 minutes) and blocked with 0.1% BSA in 0.5% PBSTx at room temperature for 2 hours. Afterwards, the samples were incubated with acetylated tubulin (1:1,000, 6-11B-1, Sigma-Aldrich), glutamylated tubulin (1:400, GT335, Adipogen) or beta-catenin (1:200, Cell Signaling Technologies), HuC/D (1:150, A-21271, Invitrogen) antibodies overnight at 4°C. The next day, the larvae were washed (0.5% PBSTx, 3 × 1 hour) and subsequently incubated with the secondary antibodies (Alexa-labeled GAR555 plus and GAM IgG2b633, 1:1,000, Thermo Fisher Scientific) in fresh blocking solution overnight at 4°C. On the third day, DAPI staining (1:1,000, Thermo Fisher Scientific) was performed for 2 hours at room temperature in 0.5% PBSTx. The larvae were thoroughly washed (3 × 1 hour) in 0.5% PBSTx and then transferred in glycerol of increasing concentrations (25%, 50%, and 75%) before being mounted in 75% glycerol. Confocal imaging was performed with Zeiss LSM 880 Axio Examiner Z1 with a Zeiss 20X plan NA 0.8 objective. For a detailed protocol, see [91].

The same fixing, staining, and imaging procedure was used on 5 dpf *foxj1b*:*GFP* larvae. These samples were incubated with the glutamylated tubulin (1:400, GT335, Adipogen) and anti-GFP-488 (1:1,000, Thermo Fisher Scientific) antibodies and Alexa-labeled 555 plus (1:500, Thermo Fisher Scientific) secondary antibody.

### Immunostaining of the adult OB

For staining the OB of *foxj1b*:*GFP* fish, brain samples were dissected and fixed using 4% PFA in PBS and incubated at 4°C overnight. The next day, the samples were washed with 0.25% PBSTx (3 × 10 minutes). The samples were then incubated with 0.05% Trypsin-EDTA on ice for 40 minutes. After incubation, they were quickly washed twice and then for 10 minutes with 0.25% PBSTx. The samples were blocked with 2% DMSO+1% BSA made in 0.25% PBSTx for 4 hours. Samples were then incubated with mouse anti-SV2 monoclonal antibody (1:1,000, DSHB), 1 ml per tube and agitated for 3 to 5 days at 4°C. The next day, the samples were washed with 0.25% PBSTx (3 × 1 hour). The samples were then incubated with secondary antibody diluted in 0.25% PBSTx, Alexa-labeled GAM555 plus (1:500, Thermo Fisher Scientific) + anti-GFP-488 (1:500, Thermo Fisher Scientific) + DAPI (1:1,000, Thermo Fisher Scientific). This was incubated for 3 days at 4°C. The stained samples were washed with 0.25% PBSTx (3 × 1 hour) and were subsequently treated with increasing concentrations of glycerol (25%, 50% for 1 hour each). The stained samples were stored at 4°C and imaged using a Zeiss Examiner Z1 confocal microscope with a Zeiss 20× plan NA 0.8 objective.

### In vivo confocal imaging of larval zebrafish

The in vivo imaging of *foxj1a*:*RFP; foxj1b*:*GFP* larvae were performed at 5 dpf. The larvae were immobilized using tricaine methanesulfonate (MS222) and were subsequently embedded in

1.5% LMP agarose (Thermo Fisher Scientific), with the nose pointing up. After the agarose was solidified, confocal imaging was performed with Zeiss LSM 880 Axio Examiner Z1 with a Zeiss 20X plan water objective, NA 1. In vivo imaging of *foxj1b:GFP; OMP:gal4; UAS:NTR-mCherry* larvae was done similarly at 4 dpf.

### Quantification of nose morphology

Confocal images of 4 dpf larval noses were investigated with Fiji/ImageJ, and their nose and nasal cavity size were measured. MCCs were counted making use of the acetylated tubulin and beta catenin immunostaining. All quantification was done on both nostrils of each animal, and the average value across nostrils per larvae was reported. Statistical analyses were done using MATLAB. The two-sample *t* test was used for analysis. $P < 0.05$ was considered as statistically significant.

### Single-cell RNA sequencing analysis

Three publicly available scRNAseq datasets of the OE of different organisms, zebrafish [57], mouse [58] and human [59], were selected and analyzed with Seurat (3.2.0) [92–95]. The datasets consisted of 4,561, 39,985, and 27,272 cells, respectively. Each dataset was clustered to identify main cell types using UMAP dimensionality reduction. Dims = 30 and resolution = 0.8 were used to form the clusters. For zebrafish, 21 cell clusters were obtained as a result. The following markers were used to identify main cell clusters: Neuron (*elavl3*), OMP-positive cells (*ompb*), Trpc2b-positive cells (*trpc2b*), sustentacular cells (*krt5*, *lamc2*, *col1a1b*), neural progenitors (*mki67*), immune cells (*mpeg1.1*, *srgn*), MCCs (*dnah5l*), supporting cells (*cfd*, *ccl25a*, *itga6b*), blood cells (*hbba1*). Clusters 17, 18, and 19 could not be identified.

The Foxj1 target genes were identified as reported in our previous work [54]. Expression levels of selected Foxj1 target genes were identified in the cell clusters and were compared in the OSN and MCC clusters.

### RNA sequencing

To isolate RNA for sequencing, 4 dpf larvae were collected in a 1.5-ml tube and placed on ice. To lyse the samples, 500 μl trizol was added and the samples were homogenized through a 27-gauge needle until the mixture looked uniform. After adding another 500 μl trizol, the samples were incubated for 5 minutes at room temperature. The larvae were then treated with 200 μl chloroform, and the tube was rocked for 15 seconds to mix the contents. The tubes were incubated for 2 minutes at room temperature and then centrifuged for 15 minutes at 12,000 rpm at a temperature of 4°C. After centrifugation, the upper aqueous phase containing RNA was mixed with equal amounts of 100% ethanol and was then loaded onto an RNA spin column (Qiagen) and centrifuged for 30 seconds at 8,000 rpm. The spin column was further incubated with 700 μl of RW1 buffer and centrifuged for 30 seconds at 8,000 rpm. The spin column tubes were then placed into a new collection tube and further treated to remove any DNA contamination by washing the tubes with 350 μl of RW1 buffer followed by DNase enzyme (Qiagen) in RDD buffer (10 μl DNase+ 70 μl RDD buffer per tube) for 45 minutes at room temperature. After incubation, 350 μl of RW1 buffer was added to the tubes and centrifuged for 15 seconds at 8,000 rpm. The tubes were then treated with 500 μl RPE buffer and centrifuged for 30 seconds. This step was repeated twice, and the tubes were then centrifuged for 1 minute at 8,000 rpm to remove any residual buffer left in the column. For RNA extraction from the column, 30 μl nuclease-free water was added and incubated for 2 minutes. The tubes were then centrifuged for 1 minute at 8,000 rpm to elute the RNA. The concentration of the extracted RNA was quantified using Nanodrop, and the quality was analyzed with a

bioanalyzer. The samples were then sequenced by BGI's DNBSEQ Technology. In order to analyze the RNA sequencing data, the paired-end sequencing reads were aligned to the Zebrafish genome (GRCz11) using GSNAP [96], and read counts per gene were determined by using featureCounts (v2.0.3; [97]) using ensemble version 106. Only paired-end reads that were uniquely mapped were kept for downstream analyses. The raw read counts were used as input for DESeq2 (v1.36.0; [98]) for differentially expressed genes. Genes with abs (log2 fold change) > 2 and p-adj < 0.1 were assumed as significantly expressed genes. Datasets are available on GEO with the following accession number GSE232397. Raw files used to generate the figures are available on Mendeley Data at this link https://data.mendeley.com/datasets/2pn963jn6y.

## Hybridization chain reaction (HCR)

HCR stainings were done according to the protocol by Choi and colleagues [99], with some small modifications [91]. All HCR probes, pre-hybridization buffer (pre-hybe), amplification buffer, short hairpins, and wash buffers were purchased from Molecular Instruments. In 1.5 ml eppendorf tubes, 15 to 20 zebrafish embryos at required stages were collected and then fixed with 1 ml of 4% PFA for overnight at 4°C. Next day, embryos were washed for 2 hours with 1× PBS with buffer change every half an hour. Finally, the embryos were transferred in absolute methanol and then stored in −20°C for overnight incubation. Next day, embryos were serially rehydrated by washing with 1 ml of 75%, 50%, and 25% methanol diluted in PBS and subsequently with PBST (1× PBS with 0.1% Tween-20) washes few times for 5 minutes. For each wash, tubes were placed on a horizontal shaker. After the washes, embryos were treated with 1 ml of Proteinase K prepared in PBT (0.8 µl of 25 mg/ml Proteinase K in 1 ml PBT). Treatment time varied according to the stages of the embryos. After the treatment, embryos were quickly washed for 2 times with 1 ml of PBST, and, finally, residual Proteinase K activity was stopped completely by adding 1 ml of 4% PFA. Embryos were incubated for 20 minutes at room temperature. Embryos were then washed 3 times with PBST for 5, 15, and 25 minutes, respectively. After the final wash, embryos were incubated in 350 µl of HCR pre-hybe prewarmed at 45°C by placing the tubes in a floater and incubating in a water bath set at 45°C for 2 hours. After the incubation, 250 µl of probe solution (prepared by adding 1 µl of 2 or more HCR probes each tagged with different initiators, in 250 µl of prewarmed pre-hybe; final concentration of 2 pmol of probe) was added in each tube. Embryos were incubated in the same water bath overnight (12 to 16 hours).

Next day, extensive washing was done. Embryos were first washed with 500 µl of HCR probe wash buffer for 2 times × 10 minutes each and thereafter 2 times for 30 minutes each with 70%, 50%, and 25% wash buffer (diluted in 5× SSCT) and final one 15-minute wash with 5× SSCT. All washes done by incubating embryos in 45°C water bath. Final 2 washes of 10 minutes done with 5× SSCT at room temperature. Embryos were then incubated for an hour in 500 µl HCR amplification buffer at room temperature. This buffer has been prewarmed at room temperature once taken out from 4°C storage. Meantime, HCR hairpin RNA solution was prepared by snap-cooling the hairpins. In separate PCR tubes, 10 µl of each h1 and h2 hairpins (both 30 pmol) for a single initiator was aliquoted. Tubes were then placed in a heat block at 95°C for 1.5 minutes and immediately cooled at room temperature and incubated for 30 minutes in the dark. Both h1 and h2 were then aliquoted in 250 µl of amplification buffer and mixed well. Once embryo incubation in amplification buffer was over, hairpin mixed buffer was then added to each tube and incubated at room temperature overnight (12 to 16 hours) in the dark.

Next day, amplification buffer mix was removed, and embryos were washed 5 times for 30 minutes with 1 ml of 5× SSCT at room temperature. This was then followed by another 4

washes with 1 ml PBST buffer for 25 minutes each. To label the nuclei, 0.3 μl of DAPI (5 mg/ml) was added in the first wash with PBST. Once PBST washes were over, embryos were finally kept in 1 ml of 70% glycerol prepared in PBS and stored in −20°C. Embryos were imaged using an Olympus FLUOVIEW FV3000 upright confocal microscope.

## Whole-mount in situ hybridization (WISH)

WISH was done according to the routine protocol. Antisense-RNA probes for *ompb* was synthesized from *ompb* CDS cloned in the TOPO-TA vector, while *cnga4* probe was synthesized according to the protocol by Hua and colleagues [100]. Primer sequences for *ompb* and *cnga4* probe synthesis are mentioned in the oligonucleotide section.

## DNA isolation for zebrafish embryo genotyping

Antibody-stained or FISH-stained embryo genotyping was done in 2 steps—DNA isolation and PCR followed by agarose gel electrophoresis. Genomic DNA isolation from individual embryos or parts of embryos such as excised head or torso was done by alkaline lysis method. A single embryo or embryo head or torso was transferred to individual PCR tube containing 20 μl of PBS. PBS was then removed and 20 μl of 25 mmol of alkaline lysis solution (25 mmol of sodium hydroxide + 0.2 mmol EDTA, prepared in sterile water) was added in the tube. The tube was transferred to a thermocycler for incubation at 95°C for 45 minutes. After incubation, the tube was taken out, vortexed, and quick spun in a table-top spinner. About 20 μl of neutralization buffer (40 mmol of Tris-HCl prepared in sterile water (pH 8.0)) was then added to the tube and mixed well. The DNA mixture was now ready for PCR. Around 1 μl of this DNA mix was used for a 10 μl of PCR reaction.

For genotyping of larval zebrafish used for the nose quantification and odor–response experiments, the samples were subjected to gDNA isolation using 50 μl PCR lysis buffer (containing 10 mM tris (pH 7.5), 50 mM EDTA, 0.2% Triton X-100, and 0.1 mg/ml Proteinase K) overnight at 50°C. To stop the reaction, the samples were heated to 95°C for 10 minutes. The samples were then centrifuged at 13,000 rpm for 2 minutes. The supernatant containing gDNA was used for qPCR. For performing qPCR, 10 μl SYBR green PCR master mix (Thermo Fisher Scientific) was mixed with 0.5 μl each of forward and reverse primer and 7 μl water to make an 18-μl reaction mixture. This mixture was added to a 96-well qPCR plate (Thermo Fisher Scientific), and 2 μl of extracted gDNA were mixed with this reaction. The samples were then analyzed based on their melting curves as wild type, heterozygous, or homozygous or by further gel electrophoresis.

## Supporting information

**S1 Fig. *foxj1* expression in the OE and axonal projection of *foxj1* expressing cells into the OB in adult zebrafish.** (**A**) Schematic showing the adult zebrafish OB connected to the OE. (**B**) Confocal image of an adult zebrafish OE showing expression of *foxj1a* (*Gt(foxj1a:2A-TagRFP)*, magenta) and *foxj1b* (*Gt(foxj1b:GFP)*, green). Note that *foxj1a* is mainly expressed at the tip of the lamellas where MCCs are located. Scale bar = 50 μm. (**C, D**) Projections of *foxj1b*- (**C**, *Gt(foxj1b:GFP)*, green) and *omp*-positive OSNs (**D**, *Tg(OMP:ChR-YFP)* green) into the OB. Glomeruli are indicated by the presynaptic marker SV2. Note that *foxj1b*-expressing OSNs project to more glomeruli than *omp*-expressing OSNs. Scale-bars = 20 μm. a: anterior, p: posterior. Tel = Telencephalon. MCC, motile multiciliated cell; OB, olfactory bulb; OE, olfactory epithelium; OSN, olfactory sensory neuron.
(TIF)

**S2 Fig. Foxj1 expression in the mouse OE at different ages.** (**A**-**C**) Immunostaining of Foxj1 at different animal ages (newborn P0 (**A**), day 5 P5 (**B**), and adult P30 (**C**)) in OSNs of wild-type mice. Border between the OE and the RE is marked by a dashed line. Brightly labeled cells in the RE are respiratory MCCs. Scale bars = 50 μm. MCC, motile multiciliated cell; OE, olfactory epithelium; OSN, olfactory sensory neuron; RE, respiratory epithelium.
(TIF)

**S3 Fig. Loss of Foxj1 results neutrophil infiltration into the nasal cavity and defective glomeruli.** (**A**, **A'**) Zoomed out view of the nasal cavity of the WT and *Foxj1*−/− mouse. Nasal cavities are denoted by asterisks. (**B**, **B'**) Level of apoptosis in the OE was similarly low in the WT (**B**) and *Foxj1*−/− (**B'**) mouse (108 ± 24, $n$ = 6, WT; 83 ± 7 cells per mm$^s$, $n$ = 6, KO; 3 mice, $p$ = 0.554) as determined by immunostaining for cleaved Caspase 3 (arrows). (**C**, **C'**) OMP immunostaining showed larger, strongly OMP-expressing glomeruli in OB of WT (**C**, top-left), but irregular, smaller glomeruli in *Foxj1*−/− mouse (**C'**, bottom-left). Wandering axons present in OB of Foxj1−/− mouse as axons did not fully converge within glomeruli overshooting in internal layers of the OB (**C'**, bottom-left, arrows). Similarly, TH expression denoting neural activity reduced in Foxj1−/− mouse (**C'**, bottom-right) showing weaker TH immunofluorescence (185.6 ± 19.7 a.u., $n$ = 14, WT; 63.6 ± 7.9 a.u., $n$ = 37, KO, 3 mice, $p$ < 0.0001) when compared to WT mouse (**C**, top-right). (**D**) Nasal cavity of the *Foxj1*−/− mouse contained neutrophils as revealed by HE stain. Scale bar = 1,000 μm (**A**, **A'**), 100 μm (**B**, **B'**, **D**), and 10 μm (**D**, inset). HE, hematoxylin–eosin; KO, knockout; OB, olfactory bulb; OE, olfactory epithelium; OMP, olfactory marker protein; TH, tyrosine hydroxylase; WT, wild type.
(TIF)

**S4 Fig. Transcriptional investigation of the OE across species.** (**A**-**A''**) Dimensionality reduction plots of detected cell clusters on single-cell RNA sequencing data from zebrafish, mouse, and human olfactory epithelial tissue (calculated by umap). Cell clusters corresponding to MCCs and OSNs are shown. (**B**-**B''**) Marker genes *omp*, *dnah5*, and *foxj1* expression across cell clusters. (**C**) Dot-plot showing expression levels of *foxj1* target genes of interest in MCC and OSN cell clusters across species. MCC, motile multiciliated cell; OE, olfactory epithelium; OSN, olfactory sensory neuron.
(TIF)

**S1 Video. Motile OSN cilia of the leopard frog *Rana pipiens*.** An isolated OSN from the amphibian species *Rana pipiens* investigated under transmission light microscopy. Movement of the OSN cilia can be clearly observed. For details on how the cell suspension was achieved, see [101].
(MP4)

**S1 Table. List of differentially expressed genes identified through RNA sequencing from 4-day-old *foxj1a* or *foxj1b* mutant zebrafish larvae ($n$ = 3 control and mutant for each genotype).**
(XLSX)

**S2 Table. List of various materials and reagents used in the study.**
(DOCX)

## Acknowledgments

We would like to thank S. Kleene, University of Cincinnati, for the recording of ciliary beating on the dissociated frog OSN shown in S1 Video; Y. Yoshihara, Riken Center for Brain Science,

for sharing the Gαolf antibody; the Fluorescent Reporter Zebrafish Cooperation Center (FRZCC), Korea, for providing FRZCC #1100; C. Wee for providing *Tg(trpc2b:gal4)* and *Tg (UAS:NTR-mCherry)* zebrafish strains and the fish facility support staff in our respective institutes for zebrafish maintenance.

## Author Contributions

**Conceptualization:** Nathalie Jurisch-Yaksi, Sudipto Roy.

**Formal analysis:** Dheeraj Rayamajhi, Mert Ege, Kirill Ukhanov, Christa Ringers, Yiliu Zhang, Summer Shijia Li, Mehmet Ilyas Cosacak.

**Funding acquisition:** Jeffrey R. Martens, Steven L. Brody, Nathalie Jurisch-Yaksi, Sudipto Roy.

**Investigation:** Dheeraj Rayamajhi, Mert Ege, Kirill Ukhanov, Christa Ringers, Yiliu Zhang, Inyoung Jung, Percival P. D'Gama, Summer Shijia Li.

**Supervision:** Caghan Kizil, Hae-Chul Park, Emre Yaksi, Jeffrey R. Martens, Steven L. Brody, Nathalie Jurisch-Yaksi, Sudipto Roy.

**Visualization:** Dheeraj Rayamajhi, Mert Ege, Kirill Ukhanov.

**Writing – original draft:** Mert Ege, Kirill Ukhanov, Nathalie Jurisch-Yaksi, Sudipto Roy.

**Writing – review & editing:** Dheeraj Rayamajhi, Mert Ege, Kirill Ukhanov, Christa Ringers, Yiliu Zhang, Inyoung Jung, Percival P. D'Gama, Summer Shijia Li, Mehmet Ilyas Cosacak, Caghan Kizil, Hae-Chul Park, Emre Yaksi, Jeffrey R. Martens, Steven L. Brody, Nathalie Jurisch-Yaksi, Sudipto Roy.

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
