## [Editor Report · Decision Letter 0]

3 Jul 2023

Dear Dr Roy, 

Thank you for submitting your manuscript entitled "Vertebrate olfactory cilia and sensory neuron differentiation is programmed by the forkhead domain transcription factor Foxj1" for consideration as a Research Article by PLOS Biology.

Your manuscript has now been evaluated by the PLOS Biology editorial staff [as well as by an academic editor with relevant expertise and I am writing to let you know that we would like to send your submission out for external peer review. However, we would like to consider it as a Short Report, thus please select that type of article from the dropdown menu when you submit the metadata (see below). Also, please note that this type of articles only has 4 main figures, so you will have to either combine some of the figures or convert some into Supplementary ones.

Before we can send your manuscript to reviewers, we need you to complete your submission by providing the metadata that is required for full assessment. To this end, please login to Editorial Manager where you will find the paper in the 'Submissions Needing Revisions' folder on your homepage. Please click 'Revise Submission' from the Action Links and complete all additional questions in the submission questionnaire.

Once your full submission is complete, your paper will undergo a series of checks in preparation for peer review. After your manuscript has passed the checks it will be sent out for review. To provide the metadata for your submission, please Login to Editorial Manager (https://www.editorialmanager.com/pbiology) within two working days, i.e. by Jul 05 2023 11:59PM.

Kind regards,

Ines

--

Ines Alvarez-Garcia, PhD

Senior Editor

PLOS Biology

---

## [Decision Letter · Decision Letter 1]

12 Sep 2023

Dear Dr Roy,

Thank you for your patience while your manuscript entitled "Vertebrate olfactory cilia and sensory neuron differentiation is programmed by the forkhead domain transcription factor Foxj1" went through peer-review at PLOS Biology. Please also accept my apologies for the delay in providing you with our decision. In this case, your article was evaluated by the PLOS Biology editors, an academic editor with relevant expertise and we consulted three independent reviewers. However, we have to date only received reports from two of them; we will forward you the third one if it is sent to us belatedly.

The reviews are attached below. As you will see, the reviewers find the conclusions novel and interesting, but they also raise several points that should be clarified. Reviewer 1 asks for a secondary marker for glomeruli to confirm that regions lacking OMP are actually glomeruli, along several clarifications. Reviewer 2 would like you to explain better the differences between the olfactory systems of zebrafish and mice in the introduction, to speculate about the identity of the additional foxj1b expressing ORNs and adding missing stats and quantifications to some of the experiments.

In light of the reviews, we are pleased to offer you the opportunity to address the reviewers' comments in a revision that we anticipate should not take you very long. We will then assess your revised manuscript and your response to the reviewers' comments with our Academic Editor aiming to avoid further rounds of peer-review, although might need to consult with the reviewers, depending on the nature of the revisions.

**IMPORTANT - SUBMITTING YOUR REVISION**

3. Resubmission Checklist

a) *PLOS Data Policy*

b) *Published Peer Review*

d) *Blurb*

Please also provide a blurb which (if accepted) will be included in our weekly and monthly Electronic Table of Contents, sent out to readers of PLOS Biology, and may be used to promote your article in social media. The blurb should be about 30-40 words long and is subject to editorial changes. It should, without exaggeration, entice people to read your manuscript. It should not be redundant with the title and should not contain acronyms or abbreviations. For examples, view our author guidelines: https://journals.plos.org/plosbiology/s/revising-your-manuscript#loc-blurb

Sincerely,

Ines

--

Ines Alvarez-Garcia, PhD

Senior Editor

PLOS Biology

Reviewers' comments

Rev. 1:

The submitted manuscript explores a role for the transcription factor Foxj1 in olfactory sensory neuron cilia development in both zebrafish and mouse. The authors demonstrate a role for this protein in non-motile ciliated cells. They effectively demonstrate changes in OSN number in the mouse OE in Foxj1 knockouts, although claims that Foxj1 controls OSN-sepcific genes are not strongly backed up in the mouse. The reduction in both iOSNs and mOSNs, without increase in apoptotic cells, and decrease in Ki67 (proliferating) cells, raises questions of if Foxj1 is more functional in the early generation of OSNs. Single cell analysis of mOSNs from WT and KO mice would strengthen claims. Additionally the bulb phenotype in mouse would largely appear to be due the reduction in OSNs, which may show more wandering due to lower activity, though there is not sufficient data to fully support that. Functional studies in zebrafish strengthen the manuscript however, and provide compelling data the mOSN function in that species is altered specifically. On the whole, the manuscript is largely well written, and the results are interesting. This manuscript would add to our understanding of cilia development in OSNs which are critical for a functional olfactory system.

Some issues needing clarrifciation

1) Figure 1: What ages are representative images from in panels L, M and N?

2) Figure 2: Not clear in the panels/legend what age the mouse sections are from. Schematics are where slices come from would be helpful. For example, images in F are clearly from different regions of the OE. In H, representative glomeruli images come from a more dorsal or lateral (WT) side and medial (KO) edges. Glomeruli have different sizes across regions of the bulb, showing images from the same region would allow better comparisons.

3) In the figure legends, it is not always clear how many animals were used for different immunolabeling experiments. The n appears to related to slides/sections, glomeruli counted and animals depending on which legend, however number of animals is not consistently reported.

4) This statement should be reworded "In case of mice, loss of Foxj1 in P21 pups resulted in a severe disfigurement of the olfactory system at the gross anatomical level, manifest in largely undeveloped turbinates and the entire nasal cavity was filled with DAPI-positive cellular masses composed of neutrophils

(S3A, SA' and SD Figs)." As written is suggests that Foxj1 was selectively targeted in P21 mice, and not analysis of Foxj1 loss on OE anatomy using P21 aged mice.

5) The statement that "aberrant innervation also resulted in smaller size of glomeruli" is not supported by data. Smaller size is most likely linked to significant reduction in both immature and mature OSNs.

6) A secondary marker for glomeruli (GAP43, or M/T dendrite marker) should be used to confirm that regions lacking OMP are actually glomeruli.

Rev. 2:

In their manuscript entitled "Vertebrate olfactory cilia and sensory neuron differentiation is programmed by the forkhead domain transcription factor Foxj1", Dheeraj Rayamajhi and coworkers show that the forkhead domain-containing transcription factor (FoxJ1), known to be involved in the activation of ciliome genes in motile cilia cell-types, is also involved in the biogenesis of immotile olfactory cilia and the development of the olfactory receptor neurons (ORNs).

To obtain their results, the authors employed whole-mount in-situ hybridizations, hybridization chain reactions, RNA sequencing techniques, In-vivo confocal imaging, immunostainings, etc.

The study is very well conducted, the logic of the experiments is fine, the methodology is sound, the statistical analysis is well done, and the authors provide convincing evidence for all their claims.

The results are extremely interesting for individuals working in the field of olfaction but also for people working in the field of ciliary physiology and for biologists in general. The obtained results are, without a doubt, a significant advance in the relevant fields. It was a pleasure to read the manuscript.

However, I feel that before publication, a few points (see below) must be clarified and better explained. I think it is necessary to discuss some of the obtained results better (see my comments below).

Specific critique:

In the Introduction, you should better explain the differences between the olfactory systems of zebrafish, mice (and humans). You do not mention that several different ORN-types, ciliated and non-ciliated, exist in zebrafish. Adding this information will help the readers to understand the study better.

In Figure 1 E and Figure H-J: Not all foxj1b-expressing cilia/cells are Gαolf-positive. Could you speculate about the identity of the additional foxj1b expressing (ciliated) ORNs? Also, is it known which morphologic ORN-type innervates the ventral medial glomeruli? I did not find this information in the manuscript.

Figure 1 M: There are still some puncta in this figure. What could this puncta possibly be?

Figure 2A: I do not see a big difference between the stainings in foxj1b mutant and foxj1a/b double mutant larvae. In the text that describes these results, you claim that there is a difference. Did you do any statistics? Could you please explain what you mean here?

In Figure 2B, you show only a small part of the OB. Why do you not show the whole OB? How is the innervation (Gαolf-positive ORNs) in the rest of the OB?

Figures 2E, F, G: How can you tell that in mutant mice, the apical ciliary layer of the OE was disrupted entirely? I cannot see/identify cilia in the wild-type mice in the figures. Also, how do you explain that some ORNs are still present in the OE of mutant mice? How can these ORNs be formed? How do these ORNs differ from those that are not formed? Better discuss.

Figure 2H: Some glomeruli completely lacked innervation by OMP-positive ORN-axons. You state that this suggests aberrant targeting by the ORNs. Why aberrant targeting? Aren't these glomeruli "empty" because the ORNs that innervate them are missing in the mutants? Did you do any quantification of the glomeruli data?

Figure 3A: You should state in the results that describe these data that these data come from wild-type animals.

Odorant-stimulation data: You show that in foxj1b mutant zebrafish, but not in foxj1a mutants, the formation of olfactory cilia is strongly reduced (Figure 2A). In Figure 2A, I do not see an apparent difference in the the ciliary signal between foxj1b and foxj1a/foxj1b mutants (see also my comment above). How does this fit together with your odorant stimulation data? How can the responses to bile acids in foxj1b mutants not be impaired but be impaired in double mutant zebrafish larvae? Please better explain and discuss this observation.

---

## [Editor Report · Decision Letter 2]

22 Nov 2023

Dear Dr Roy,

Thank you for your patience while we considered your revised manuscript entitled "Vertebrate olfactory cilia and sensory neuron differentiation is programmed by the forkhead domain transcription factor Foxj1" for publication as a Short Report at PLOS Biology. This revised version of your manuscript has been evaluated by the PLOS Biology editors and by the Academic Editor.

Based on our Academic Editor's assessment of your revision, we are likely to accept this manuscript for publication, provided you satisfactorily address the data and other policy-related requests stated below.

In addition, we would like you to consider a suggestion to improve the title:

"The forkhead transcription factor Foxj1 controls vertebrate olfactory cilia biogenesis and sensory neuron differentiation" 

We expect to receive your revised manuscript within two weeks. 

*Published Peer Review History*

*Press*

Sincerely,

Ines

--

Ines Alvarez-Garcia, PhD

Senior Editor

PLOS Biology

DATA POLICY: IMPORTANT - PLEASE READ CAREFULLY

Fig. 1O; Fig. 2D; Fig. 3D, D’ and Fig. 4D-D’’’’, E-E’’’’

***Please also make the data you have deposited in GEO (GSE232397) publicly available at this stage.

---

## [Editor Report · Decision Letter 3]

12 Dec 2023

Dear Dr Roy,

Thank you for the submission of your revised Short Report entitled "The forkhead transcription factor Foxj1 controls vertebrate olfactory cilia biogenesis and sensory neuron differentiation" for publication in PLOS Biology. On behalf of my colleagues and the Academic Editor, Piali Sengupta, I am delighted to let you know that we can in principle accept your manuscript for publication, provided you address any remaining formatting and reporting issues. These will be detailed in an email you should receive within 2-3 business days from our colleagues in the journal operations team; no action is required from you until then. Please note that we will not be able to formally accept your manuscript and schedule it for publication until you have completed any requested changes.

PRESS

Sincerely, 

Ines

--

Ines Alvarez-Garcia, PhD

Senior Editor

PLOS Biology
